# The Effects of Environmental Changes on Plant Species and Forest Dependent Communities in the Amazon Region

Diego Oliveira Brandão [1],*, Lauro Euclides Soares Barata [2] and Carlos Afonso Nobre [1]

1 Postgraduate in Earth System Science (PGCST), National Institute for Space Research (INPE), Sao Jose dos Campos 12227-010, Brazil; cnobre.res@gmail.com
2 R&D Laboratory on Bioactive Natural Products, Federal University of West Para (UFOPA), Santarem 68035-110, Brazil; lauroesbarata@gmail.com
* Correspondence: diegobrandao779@gmail.com

**Abstract:** We review the consequences of environmental changes caused by human activities on forest products and forest-dependent communities in the Amazon region—the vast Amazonas River basin and the Guiana Shield in South America. We used the 2018 and 2021 Intergovernmental Panel on Climate Change reports and recent scientific studies to present evidence and hypotheses for changes in the ecosystem productivity and geographical distribution of plants species. We have identified species associated with highly employed forest products exhibiting reducing populations, mainly linked with deforestation and selective logging. Changes in species composition along with a decline of valuable species have been observed in the eastern, central, and southern regions of the Brazilian Amazon, suggesting accelerated biodiversity loss. Over 1 billion native trees and palms are being lost every two years, causing economic losses estimated between US$1–17 billion. A decrease in native plant species can be abrupt and both temporary or persistent for over 20 years, leading to reduced economic opportunities for forest-dependent communities. Science and technology investments are considered promising in implementing agroforestry systems recovering deforested and degraded lands, which could engage companies that use forest products due to supply chain advantages.

**Keywords:** agroforestry system; climate change; deforestation; forest degradation; non-timber forest products

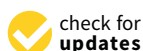

## 1. Introduction

Environmental changes caused by human activities alter the water, energy, and carbon cycles in the Amazon region [1–3]. This has resulted in biological changes across several plant species [4–6], some of which are used in both regional and global trade and represent important sources of food and income for people [7,8]. Reports from local people and scientific studies point to the effects of deforestation, forest degradation, and climate change on native plant species [9–11]. Indeed, people who are typically dependent on natural resources and ecosystem services are the most threatened by plant species productivity and geographical distribution changes [12]. However, there is a lack of scientific literature concerning the effects of environmental changes on plant species and forest-dependent communities in the Amazon region [13–15].

The change in plant species composition in degraded areas is well established by studies conducted in the Amazon and includes increased mortality of wet-affiliated stems and increased small-seeded plant species [16,17]. In addition, droughts have become more intense in the region [18,19], and the ensuing regional hydrological cycle changes, in turn, alter plant ecophysiology and ecosystem productivity [20–22]. In the context of climate change [2], ecological niche modeling has estimated a decrease in the geographical distribution of native species that are highly embedded in the region's economy [23]. For example, the area of occurrence of the *Bertholletia excelsa* Bonpl. (commonly known as

Brazil nut) could decrease by 25% by 2050, affecting 2239 extractive families of non-timber forest products (NTFP) who live in protected areas [23]. This may, in turn, potentially result in lower wood and seed density [17], as changes in species composition are reducing the density of trees and palms widely employed by local populations [8].

We aimed to broadly characterize environmental changes in the Amazon to assess why the livelihoods of people who depend on forest products are increasingly threatened. We specifically review the effects of deforestation, forest fragmentation, selective logging, forest fire, droughts, global warming, and possible "savannization" on both the productivity and geographical distribution of plant species. To exemplify this, we prepared a list of 30 native plant species negatively impacted by environmental changes, alongside a list of 30 native species that thrive in deforested and degraded lands. Maps of the distribution of typical savannah species found in the Brazilian Amazon are presented for nine species. We also discuss estimates of the annual number of trees and palms that are being destroyed by deforestation in the Amazon and potential economic losses. The conclusions are multidisciplinary and combine botany, ecology, economy, forestry, and sustainability science, and will offer an overview of recent trends and highlight future strategies for improving agroforestry systems.

It is important to understand the interactions between environmental changes and plant species to collectively create strategies to drive a future agroforestry system agenda focused on people, business, and forest restoration. The plant diversity in the Amazon is the highest in the world, with an estimated 15,000 trees and palms species living alongside 30 million people [24]. The Amazon rainforest is located in South America, spread over 6.7 million square kilometers ($km^2$), encompassing Brazil, Bolivia, Colombia, Ecuador, French Guiana, Guyana, Peru, Suriname, and Venezuela. Hence, such studies are crucial for understanding the rapid land degradation currently taking place in the Amazon region and the consequences for social and economic activities [13,25,26].

## 2. Materials and Methods

### 2.1. Criteria for Literature Selection

This literature review was prepared in accordance with the Intergovernmental Panel on Climate Change [2,3], scientific studies on anthropogenic drivers of environmental changes, and technical studies in the Amazon region. As feasible as possible, we present all literature relevant to understand how deforestation, forest degradation (fragmentation, selective logging, and forest fires), and climate change (droughts, global warming, and "savannization") affect the productivity and geographical distribution of plant species in the largest rainforest in the world. Most results and discussion are presented employing recent literature in an attempt to link ecological and economic regards concerning the dynamic interaction between people and forest ecosystems in the Amazon. Finally, we have made an effort to indicate opinions consistent with available data and recent discourses.

2.1.1. Building Tables to Demonstrate Reducing and Increasing Populations of Plant Species in Deforested Amazon Lands

Many articles presented in this review were included because they provide empirical examples of reducing and increasing populations of plant species in deforested and abandoned areas. The decreasing populations of plant species are also characterized by being widely employed by local populations (i.e., Indigenous Peoples, riverine communities, Amazonian mestizos, and smallholder farmers), including for income generation by using fruits, seeds, resins, and other NTFPs. On the other hand, plant species exhibiting increasing populations are characterized by being much less used by local populations, as their forest products are not sold or bought in the region. The increases in these plant populations have taken place even without the management of these species in deforested and abandoned lands. In addition to scientific articles, the distinction between plant species in the two tables is made according to the authors' experience with local plant species. Thus, 30 species are presented in each table to exemplify changes in species composition

that tend to reduce the livelihoods of forest-dependent communities in the Amazon. The tables presenting increasing and decreasing species populations are presented with their respective botanical family and scientific name information according to the Angiosperm Phylogeny Group (APG IV) and scientific references.

### 2.1.2. Maps Concerning Typical Amazon Savannah Species

The representation of typical savannah species found in the Brazilian Amazon was carried out in map form according to scientific studies and speciesLink network database research. Scientific studies provided information on species composition in Amazon savannah areas. The speciesLink network database searches were carried out according to the scientific names of nine selected species to confirm their distributions as mainly in the savannah regions present in central Brazil and northern South America. Finally, the maps presenting the georeferenced occurrence points of the nine species were grouped together to produce the figure resulting from the combination of scientific studies and speciesLink network database research.

### 2.1.3. Annual Range of Economic Importance Losses Caused by Deforestation

We employed peer-review articles to estimate the annual range of economic losses caused by Amazon deforestation [27–29]. The net annual revenues from NTFP were used to estimate a minor set of ecosystem services losses due to deforestation [27]. We considered the annual revenues from NTFP in US$422 ha$^{-1}$ year$^{-1}$, according to estimates for the Peruvian Amazon [27]. The average monetary values from tropical forests were used to estimate a major set of ecosystem services losses by deforestation [28]. We considered the average monetary value estimated as US$5264 ha$^{-1}$ year$^{-1}$, according to calculations based on a total of 17 types of ecosystem services, including provisioning services (i.e., food, water, raw materials, genetic and medicinal resources), regulating services (i.e., air quality regulation, climate regulation, erosion prevention, biological control), habitat services (i.e., nursery service and genetic diversity), and cultural services (i.e., recreation) [28]. The annual average Amazon region deforestation estimated as in 16,686 km$^2$ during 2002–2018 was used [29]. Thus, we present and discuss an annual range of losses of economic importance caused by deforestation in the Amazon.

### 2.1.4. Potential for Agroforestry System Strategy Improvement

We also discuss the potential of agroforestry systems in reducing the negative impacts of environmental changes on forest-dependent communities. The main native and exotic plant species present in agroforestry systems in the Amazon are presented, as well as their implementation costs and financial indicators. Furthermore, we discuss the potential use of typical savannah species in agroforestry Amazon systems and the importance of a research agenda on plant species that thrive on deforested and abandoned lands.

## 3. Results

### 3.1. Deforestation Has Significantly Reduced Both Seed and Fruit Production and Altered Species Composition

Deforestation refers to the conversion of a forest to another form of land cover, such as agriculture croplands, pastures, or permanent infrastructure works [13], or a decrease in tree cover to below 10% for a long time period [30]. Deforestation is the most abrupt form of land-use change in the Amazon region. Deforestation and forest degradation, such as fragmentation, selective logging, and forest fires, leads to decreased forest capacity in providing both forest products and services [5,13], including food, water, raw materials, medicinal resources, climate regulation, pollination, and nutrient cycling [8,31,32]. Deforestation has reduced evapotranspiration and increased surface temperature by degraded lands [33], leading to changes in species composition in the most modified regions [10,16,34,35].

Deforestation in the Brazilian Amazon fuels the global economy and is enabled by weakening environmental governance [13,36,37]. Deforestation is concentrated in areas

that are home to highways, hydropower generation, mineral and wood extraction companies, and animal and vegetable protein production farms [13,38,39]. Satellite image assessments of the Brazilian Amazon identified an annual average deforestation rate of 13,856 $\pm$ 6578 km$^2$ (shallow forest cuts) from 1988 to 2020, with a 29,059 km$^2$ peak in 1995 [40]. However, the current rate of forest degradation is even higher, with 60% associated with fragmentation and edge effects and 40% linked to selective logging and forest fires [41]. Amazonian forest degradation between 1995 and 2017 was estimated at 17% for the entire forest region, the equivalent of 1,036,800 $\pm$ 24,800 km$^2$ [42].

Accumulated Brazilian Amazon deforestation has reduced the forest biomass in over 813,485 km$^2$ [43]. The average density of trees and palms in the Amazon with trunks $\geq$10 cm in diameter measured at 1.3 m from the ground is 520 stems per hectare (ha$^{-1}$) [16]. Deforestation reduces aboveground live biomass by an average of 255 to 53 tons/ha$^{-1}$ [44]. These studies and the annual average deforestation (an estimated 13,856 km$^2$) reveal the number of tree and palm reductions ranging between 576,409,600 and 648,460,800 year$^{-1}$ in the Brazilian Amazon alone [16,40,44]. In turn, the number of municipalities in the Brazilian Amazon carrying out NTFP extraction and native plant cultivation in agroforestry systems declined between 22 and 38%, which is associated with deforestation [8].

Deforestation has also led to changes in plant species composition, thereby reducing the availability of economically important NTFP in the Amazon region. The NTFP acai berry (*Euterpe precatoria* M.), andiroba (*Carapa guianensis* Aubl.), and tonka-bean (*Dipteryx odorata* (Aubl.) Willd.) are empirical examples of reduced harvests with increasing deforestation [8]. Species composition in deforested and abandoned areas depends on land-use history (i.e., soybean crop or cattle raising), propagule dispersal, and seed predation [45]. The presence or absence of certain plant species also depends on resistance to herbivory and competition and facilitated interactions between native and invasive species, as well as management practices such as regenerating and colonizing vegetation [45]. Adaptations to soil compaction and loss of both soil fertility and organic matter are significant for species performance following deforestation and abandonment [45,46]. Species from the genera *Cecropia* and *Vismia* are increasing in degraded lands [16,34], although they are not yet being used as NTFP [8].

The highest rate of deforestation occurs in an area known as the Brazilian Arc of Deforestation, located mainly on the eastern and southern edges of the Amazon region [47–49]. Due to the region's natural climatic variability, it has a lower mean annual rainfall rate and a greater temperature range [13]. Furthermore, this region has been identified as more vulnerable to climate change due to increased deforestation, global warming, and forest fires compared to the northern and western Amazon regions [19,39,50]. Hence, forest-dependent communities are most impacted by deforestation effects, mainly in these regions of the Brazilian Amazon [6,8,13,23,51].

In sum, deforestation implications are negative for the people who depend on the forest. In fact, over 500 million trees and palms are being destroyed each year [8]. Changes in species composition resulting from deforestation tend to decrease the density of trees and palm most used by local populations and increase the density of less used plant species in the region [8,34]. Due to these reasons, people can obtain fewer forest products for their livelihoods. Therefore, the social and economic activities of people who depend on the forest are being significantly reduced by deforestation [8,13].

### 3.2. Forest Fragmentation Has Reduced Seed and Fruit Diversity and Density

Forest fragmentation refers to the conversion and isolation of continuous forests into smaller fragments separated by non-forest vegetation [5,52]. There are an estimated 23,491,573 forest fragments in the tropical South American region, where the Amazon Forest is located, averaging 35 hectares and with an average aboveground carbon value of 101 tons/ha$^{-1}$ [53]. Satellite image analyses of the Amazon have shown about 160,000 forest fragments ranging from 1 to 100 ha$^{-1}$ [54]. Most of the Amazon fragmentation occurs

mainly in Brazil, with the number of forest fragments multiplying manifold from 2601 in 1976 to 38,270 in 2010 [55].

Studies have observed physical vegetation structure changes in response to increased tree mortality rates after forest fragmentation. The annual mortality rate of trees over 60 cm in diameter located within 300 m of the forest edge increased by 281% after forest edge formation compared to the mortality of large trees far from non-forest vegetation [56]. Tree mortality rates range between $2.49 \pm 1.50\%$ year$^{-1}$ and $3.67 \pm 0.70\%$ year$^{-1}$ near the edge of forest fragments, higher than values observed within the forest, ranging from $1.05 \pm 0.22\%$ year$^{-1}$ to $1.23 \pm 0.43\%$ year$^{-1}$ [56,57].

The effect caused by non-forest edges intensifies NTFP decreases [8]. The higher mortality rates of large trees are due to a combination of factors, including greater wind exposure, microclimate changes (for example, lower humidity and higher temperatures [58]), and increased number of lianas [52,59]. Among large trees in the Amazon region, *B. excelsa* is the most important economically in Bolivia, Brazil, and Peru [60–62]. *B. excelsa* trees may reach between 30 and 50 m in height when mature [63], and their mortality in forest fragments has reduced seed production, mainly in the eastern forest area [64–66].

Seed dispersal is also affected by the size of the forest fragment [35]. Compared to intact forests, there is a threefold decrease in seed dispersal in 100 ha$^{-1}$ forest fragments, while there is a sixfold decrease in 1–10 ha$^{-1}$ fragments [35]. Furthermore, there is a positive association between the number of tree species and forest fragment size [52]. Thus, fragmented forests contain fewer plant and tree species per area compared to intact forests, and local extinction rates of both pollinator and animal dispersers, such as butterflies, beetles, birds, and primates, are higher in smaller fragments [31,52,67,68]. Decreases in animal species are associated with reduced seed availability and diversity in forest fragments [35]. Changes caused by forest fragmentation can persist for over 20 years [69,70].

*3.3. Selective Logging Reduces the Wood Density and Availability of Non-Timber Forest Products*

Selective logging is the clear cut of only certain tree species from a forest. An Amazonian species known as rosewood, *Aniba rosiodora* Ducke, is an example of how selective logging can affect the regional economy by reducing tree numbers [71,72]. On average, selective rosewood felling between 1945 and 1974 reached 30,000 tons/year$^{-1}$, with a 1% essential oil yield following distillation, exporting a total of about 300 tons/year$^{-1}$ of oil in the central Amazon Maués region alone [71]. Thus, selective logging led to a reduction of rosewood trees, with oil production varying from 26 to 36 tons/year$^{-1}$ in the 1990s and 2000s in a previously important region for national oil production [71]. Currently, official data indicate only 2 tons/year$^{-1}$ of essential oil extracted from rosewood in the region [73].

The practice of selectively cutting trees without replanting has reduced the commercial use of rosewood oil across the Amazon. In the 2000s, only some municipalities in the Amazon region produced rosewood oil, compared to the initial production range for this product [71,74]. Selective rosewood logging has been restricted in Brazil since 1997, and the species was added to the Convention on International Trade in Endangered Species of Wild Fauna and Flora (CITES) list [71,75]. At present, essential oil is obtained from pruning leaves and branches of cultivated trees [7,71,72,75].

People who extract NTFP report that the number of trees of medicinal value (for example, *C. guianensis* and *Hymenaea courbaril* L.) have reduced within a 200 km radius from their communities due to selective logging [9]. Some NTFPs are now obtained from trees located over 1000 km away from the consumer market [9,76,77]. Selective logging harms the poorest communities that depend on NTFPs for subsistence and income [10,25,78]. Reduction in the NTFP availability, attributed to conflicting uses with the timber sector, has also led to greater competition among families who extract them [10].

Forests that have been selectively logged are susceptible to fire, as they comprise an average of 179 tons/ha$^{-1}$ of combustible mass, such as litter and thin branches, compared to primary forests, which contain about 56 tons/ha$^{-1}$ [79]. These forests store about 88 tons

of carbon ha$^{-1}$ less than undisturbed forests in the same region [80]. Aboveground carbon reduction varies between 47 and 75% as a result of selective logging [80]. Thus, there has been a dramatic increase in the number of forest fires, burning NTFP can be either regenerated, collected, or traded [10,11], killing more trees than palms [81].

Therefore, hundreds of plant species have declined or disappeared in the Amazon as a result of selective logging. According to Martini et al., 1994, biological tree characteristics, such as the size of the occurrence area, the density of both seedlings and mature trees, bark thickness, and growth rate, can explain timber sector preferences and predict local extinction susceptibility [82]. As a result, the number of tree species and carbon stocks decreases, and the forest consequently becomes depleted in wood and NTFPs for over 20 years [83]. Some examples of plant species widely employed by local populations suffering population decreases in modified forests are displayed in Table 1.

**Table 1.** Examples of plant species suffering population decreases in deforested and degraded lands in the Amazon.

| Botanical Family | Species Name | Scientific References |
|---|---|---|
| Anacardiaceae | *Anacardium spruceanum* Benth. ex Engl. | [82] |
| Annonaceae | *Xylopia nitida* Dunal | [82] |
| Apocymaceae | *Aspidosperma album* (Vahl) Benoist ex Pichon | [82] |
| Araliaceae | *Didymopanax morototoni* (Aubl.) Dec. & Pla. | [82] |
| Arecaceae | *Euterpe oleracea* Mart. | [8] |
| Arecaceae | *Euterpe precatoria* Mart. | [8] |
| Bignoniaceae | *Handroanthus serratifolius* (Vahl) S.Grose | [82] |
| Burseraceae | *Protium tenuifolium* (Engl.) Engl. | [82] |
| Caryocaraceae | *Caryocar glabrum* (Aubl.) Pers. | [82] |
| Clusiaceae | *Caraipa grandifolia* Mart. | [82] |
| Combretaceae | *Terminalia parvifolia* (Ducke) Gere & Boatwr. | [82] |
| Dichapetalaceae | *Tapura singularis* Ducke | [82] |
| Euphorbiaceae | *Alchorneopsis floribunda* Müll.Arg. | [82] |
| Fabaceae | *Copaifera duckei* Dwyer | [82] |
| Fabaceae | *Dinizia excelsa* Ducke | [82] |
| Fabaceae | *Dipteryx odorata* (Aubl.) Willd. | [8] |
| Fabaceae | *Eperua falcata* Aubl. | [10] |
| Goupiaceae | *Goupia glabra* Aubl. | [82] |
| Humiriaceae | *Sacoglottis amazonica* Mart. | [82] |
| Lauraceae | *Aniba rosiodora* Ducke | [71] |
| Lecythidaceae | *Bertholletia excelsa* Bonpl. | [68] |
| Malpighiaceae | *Byrsonima aerugo* Sagot | [82] |
| Meliaceae | *Carapa guianensis* Aubl. | [8] |
| Moraceae | *Brosimum acutifolium* Huber | [82] |
| Olacaceae | *Minquartia guianensis* Aubl. | [82] |
| Proteaceae | *Euplassa pinnata* (Lam.) I.M.Johnst. | [82] |
| Rutaceae | *Euxylophora paraensis* Huber | [82] |
| Sapotaceae | *Pouteria pariry* (Ducke) Baehni | [82] |
| Sapotaceae | *Pouteria macrophylla* (Lam.) Eyma | [82] |
| Vochysiaceae | *Qualea coerulea* Aubl. | [82] |

### 3.4. Forest Fires Destroy Trees and Strongly Affect Species Composition

Forest fires are more frequent in the dry season and areas with a longer one [19]. Thus, southern and eastern Amazon regions are more susceptible to fires due to lower precipitation rates compared to the northern and western regions [13,29,84,85]. The vegetation in the regions is characterized as an open ombrophilous forest with an average annual rainfall of 2096 ± 309 mm [85]. In the northern and western regions, equatorial forest and white-sand vegetation are predominant, and the average rainfall rate is 2810 ± 528 mm and 2274 ± 397 mm, respectively [85].

The occurrence of forest fires has increased and is directly correlated with climate change, deforestation, degradation, and the use of fire in agriculture [36,86]. Between

2003 and 2015, forest fires in intact forest areas gradually increased from 11 to 25% [19]. In 2015, the satellite-installed MODIS sensor detected 114,558 forest fires in the Brazilian Amazon [19]. Increases in Pacific Ocean surface water temperatures lead to an increase in forest fires in the eastern Amazon [87]. On the other hand, increasing surface water temperatures of the Atlantic Ocean has resulted in more fires in the southern and southwestern Amazon regions [87].

Forest fires destroy trees and reduce species density due to injuries to plant roots, trunks, and crowns, which may result in death [88,89]. Tree mortality after fires is associated with the morphological characteristics of the affected species, such as bark thickness and xylem and phloem vessels protection [22,81,88]. Species with thinner periderms (shell) are fire intolerant, and their populations reduce in areas that suffer frequent fires [81,90]. For example, *Protium* spp. and *Pouteria* spp., which are used to produce cosmetics, perfumes, and foods, decrease following repeated fire events [91–93]. The density of *Protium* trees decreased from 69 ha$^{-1}$ in forests without fires to 15 ha$^{-1}$ after one fire and to 2 ha$^{-1}$ after two, while *Pouteria* trees decreased from 17 ha$^{-1}$ in forests without fires to 13 ha$^{-1}$ after one fire and to 0 ha$^{-1}$ after two [81].

The number of trees that bear fruit in forests affected by fires is lower than in forests without fires [81]. On the other hand, forest fires positively influence or do not cause differences in the fruiting of palms [46,90,94]. The resilience of palms to forest fires is due to the protection offered by phloem and xylem vessels against thermal stress [88]. The density of palms ranges between 0 and 19% in forest inventories, averaging 3.4% [95]. In deforested and abandoned areas, for example, the *Attalea speciosa* Mart. ex Spreng. palm represents 12% of the regenerated vegetation cover [46]. For this reason, palm density can increase in areas affected by frequent forest fires [23,46,81,94].

Finally, forest fires reduce the productivity and geographical distribution of several plant species while increasing that of a few other species, mainly in municipalities in the central-southern and eastern regions of the Brazilian Amazon. The mortality of regenerating plants and seedling banks increases between 40 and 70% after two fire events [96]. However, Mesquita et al. observed that fires tend to increase the density of several *Vismia*, such as *Vismia guianensis* (Aubl.) Choisy and *Vismia Japurensis* Reichardt [34]. In addition, to palm and *Vismia*, other native species that increase their densities in areas suffering recurrent fires include *Banara guianensis* Aubl., *Bellucia imperialis* Saldanha & Cong., *Croton diasii* Pires ex Secco & P.E.Berry [97,98]., and others listed in Table 2. All of these plant species are not of economic value yet. Therefore, forest fires threaten forest-dependent communities in the Amazon region [9–11].

**Table 2.** Examples of plant species undergoing population increases in deforested and degraded lands in the Amazon.

| Botanical Family | Species Name | Scientific References |
|---|---|---|
| Annonaceae | *Guatteria punctata* (Aubl.) R.A.Howard | [98] |
| Arecaceae | *Attalea speciosa* Mart. ex Spreng. | [46] |
| Cannabaceae | *Trema micrantha* (L.) Blume | [97] |
| Dilleniaceae | *Curatella americana* L. | [46] |
| Euphobiaceae | *Croton diasii* Pires ex Secco & P.E.Berry | [97] |
| Euphobiaceae | *Croton matourensis* Aubl. | [98] |
| Euphorbiaceae | *Sapium marmieri* Huber | [46] |
| Fabaceae | *Apuleia leiocarpa* (Vogel) J.F.Macbr. | [46] |
| Fabaceae | *Inga thibaudiana* DC. | [98] |
| Hypericaceae | *Vismia amazonica* Ewan | [34] |
| Hypericaceae | *Vismia bemerguii* M.E.Berg | [34] |
| Hypericaceae | *Vismia cauliflora* A.C.Sm. | [34] |
| Hypericaceae | *Vismia cayennensis* (Jacq.) Pers. | [34] |
| Hypericaceae | *Vismia guianensis* (Aubl.) Choisy | [34] |
| Hypericaceae | *Vismia japurensis* Reichardt | [34] |

**Table 2.** *Cont.*

| Botanical Family | Species Name | Scientific References |
|---|---|---|
| Malpighiaceae | *Byrsonima duckeana* W.R.Anderson | [98] |
| Malpighiaceae | *Byrsonima stipulacea* A.Juss. | [34] |
| Malvaceae | *Eriotheca longipedicellata* (Ducke) A.Robyns | [97] |
| Melastomataceae | *Bellucia grossularioides* (L.) Triana | [98] |
| Melastomataceae | *Bellucia imperialis* Saldanha & Cogn. | [98] |
| Rubiaceae | *Coutarea hexandra* (Jacq.) K.Schum. | [46] |
| Rutaceae | *Zanthoxylum rhoifolium* Lam. | [97] |
| Salicaceae | *Banara guianensis* Aubl. | [97] |
| Salicaceae | *Casearia decandra* Jacq. | [97] |
| Salicaceae | *Casearia sylvestris* Sw. | [46] |
| Solanaceae | *Solanum crinitum* Lam. | [97] |
| Urticaceae | *Cecropia purpurascens* C.C.Berg | [34] |
| Urticaceae | *Cecropia sciadophylla* Mart. | [98] |
| Urticaceae | *Pourouma apiculata* Spruce ex Benoist | [98] |
| Vochysiaceae | *Erisma uncinatum* Warm. | [46] |

### 3.5. Droughts Increase the Mortality of Plant Species

Droughts occur due to interannual variability in natural phenomena, such as water circulation patterns and surface temperatures of the Pacific and Atlantic Ocean [18,99]. Extreme droughts in the Amazon region occurred in 1912, 1925/26, 1964, 1982/83, 1988, 1992, 1997/98, 2005, 2007, 2010, and 2015/16 [100,101]. The 2005 Amazon drought affected an area of 1.9 million $km^2$, with its epicenter located in the southwestern region, while the 2010 drought spread to 3 million $km^2$, with its epicenter located in southwestern Amazon, north-central Bolivia, and in the Brazilian state of Mato Grosso [102]. The 2005 and 2010 droughts were correlated with an anomalous warming of the Atlantic, while the 2015–2016 events were associated with a combination of anomalous surface water warming of the tropical and northern Pacific and Atlantic Ocean regions [19]. Field measurements indicate a temporal trend of the maximum cumulative water deficit of $-1.1$ mm/year$^{-1}$ between 1985 and 2014 throughout the entire Amazon region [16].

Droughts lead to water deficits for physiological vegetation processes, resulting in gradual phenological pattern changes, reduced wood production, tree recruitment, and fruit quality, and increased tree mortality [22,89,103–105]. In effect, droughts cause leaf loss and decreased litter moisture, and increased susceptibility to forest fires [89,104,105]. As an extreme event, the combination of drought and forest fires has increased tree mortality by over 200% in the Amazon, although palm species are more resilient [81,106].

Droughts reduce sap velocity flow between 35 and 70%, directly correlated with increased atmospheric vapor pressure deficit (VPD) during drought events [20]. Decreased sap flow is a plant response to embolism and stomatal closure. Embolism refers to the formation of air bubbles in the xylem, caused by increased water tension within the plant, while air dissolved in water expands, causing blockages, reducing sap velocity flow, and increasing plant mortality [3,20].

There is a negative relationship between the length of the dry season and aboveground biomass [107], as well as between average rainfall and the number of fruiting species in the Amazon [108]. Thus, the eastern and southern Amazonian forests naturally produce lower amounts of forest products, which may decrease even further with increasing drought frequency and intensity [19,20,96,104,105]. On the other hand, the Intergovernmental Panel on Climate Change has indicated both low confidence and a strong theoretical expectation that Amazon drying and deforestation will cause rapid changes in the regional water cycle by 2100 [3].

### 3.6. Global Warming Has Led to Changes in Vegetation

The global mean surface temperature between 2011 and 2020 ranged between 1.34 and 1.83 °C higher over land compared to the period from 1850 to 1900 [3]. Temperature

increases in the Brazilian Amazon are significant across 95% of all meteorological stations, with an average annual rise of 0.04 °C between 1973 and 2013 [109]. The average temperature increase in the Amazon is estimated to reach 6 °C by 2100, with high greenhouse gas emissions [51].

Temperature is a climatic variable that influences the geographic range of plant species [2]. A temperature increase of 1.5 °C will reduce over half the geographic range of 8% of terrestrial plant species, while a 2 °C increase will affect 16% of these species [2]. The average temperature in the Amazon is currently 28 °C [21,109], and gross ecosystem productivity is known to decrease at temperatures above 27 °C [21]. Indeed, estimates for the Amazon demonstrate that 30% of all plant species and 47% of its entire geographical distribution will be reduced gradually by 2050 due to the combination between global warming and deforestation [6].

Increases in both the VPD and temperature negatively interfere with photosynthetic capacity and wood production in the Amazon [107,110] by altering hormone formation and cellular lipid and protein synthesis [111]. High VPD reduces productivity [21] by decreasing the number of flowers and shortening vegetation exposure time to pollinators [111]. Therefore, forest-dependent communities in the Amazon will be significantly affected by rapid reductions in the productivity and geographical distribution of forest products due to increased temperatures over the next few decades [4,6,13,23].

*3.7. Changes to Degraded Savanna-like Vegetation*

Possible mechanisms of future "savannization" of the Amazon have been put forth by several studies [1,50,51,112,113]. The gradual process of replacing forest vegetation with savannah-like open canopy degraded ecosystems may begin as early as in the 2020–2029 decade [51], as a result of forest deforestation rates ranging between 20 and 40%, global warming above 4 °C, and forest fires [13,114]. Deforestation causes climate change in the Amazon due to differences in albedo, surface roughness, and evapotranspiration [1,115,116], with indications that the complete replacement of forests by pastures or soybean crops can reduce annual rainfall rates by between 9 and 25% [1,116]. Significant changes such as the reduction in precipitation, an increase in temperature, and the length of the dry season have already been shown in the Amazon [16,117,118]. Despite this, floristic composition during the changes to degraded savanna-like vegetation process remains poorly studied [1,50,114].

Salazar et al. (2007) note that typical tropical and sub-tropical savannah plant species would require hundreds or thousands of years to occupy the driest and hottest Amazon regions via natural migration processes [51]. However, plant species are also dispersed by people, and many are geographically associated with human settlements [119,120]. Savannah areas are present in Santarém, Humaitá, and Vilhena, in addition to other municipalities, where typical savannah plant species are found such as *Qualea grandiflora* Mart., *Xylopia aromatica* (Lam.), and *Caryocar brasiliense* Cambess. [121–123]. Thus, typical savannah species already occur in the Amazon due to both natural dispersion and human activities (Figure 1).

There are already some areas with characteristic savannah vegetation distributed throughout the Amazon and reported both increased wet-climate species mortality rate in humid climate and alterations that increased the number of species adapted to drier climatic conditions. Miranda et al. (2006) detected *Caraipa savannarum* Kubitzki in Vilhena and reported it as the first recorded occurrence of this species in the southern Amazon [122]. In 2021, *C. savannarum* was reported in dozens of municipalities in this region (Figure 1i). It is common in savannahs in northern South America and does not occur in Brazilian savannah, indicating that it has increased its population in the southern Amazon region [122]. Esquivel-Muelbert and co-authors (2019) reported a recruitment rate of 3.4% year$^{-1}$ in the Amazon of species belonging to the *Cecropia* genus, which is common in disturbed Amazonian lands and central Brazilian savannahs [16,34]. The presence of *C. savannarum*

and increasing *Cecropia* recruitment rates suggest that the local species composition has become adapted to drier environmental conditions [1,16,17,34,51,122].

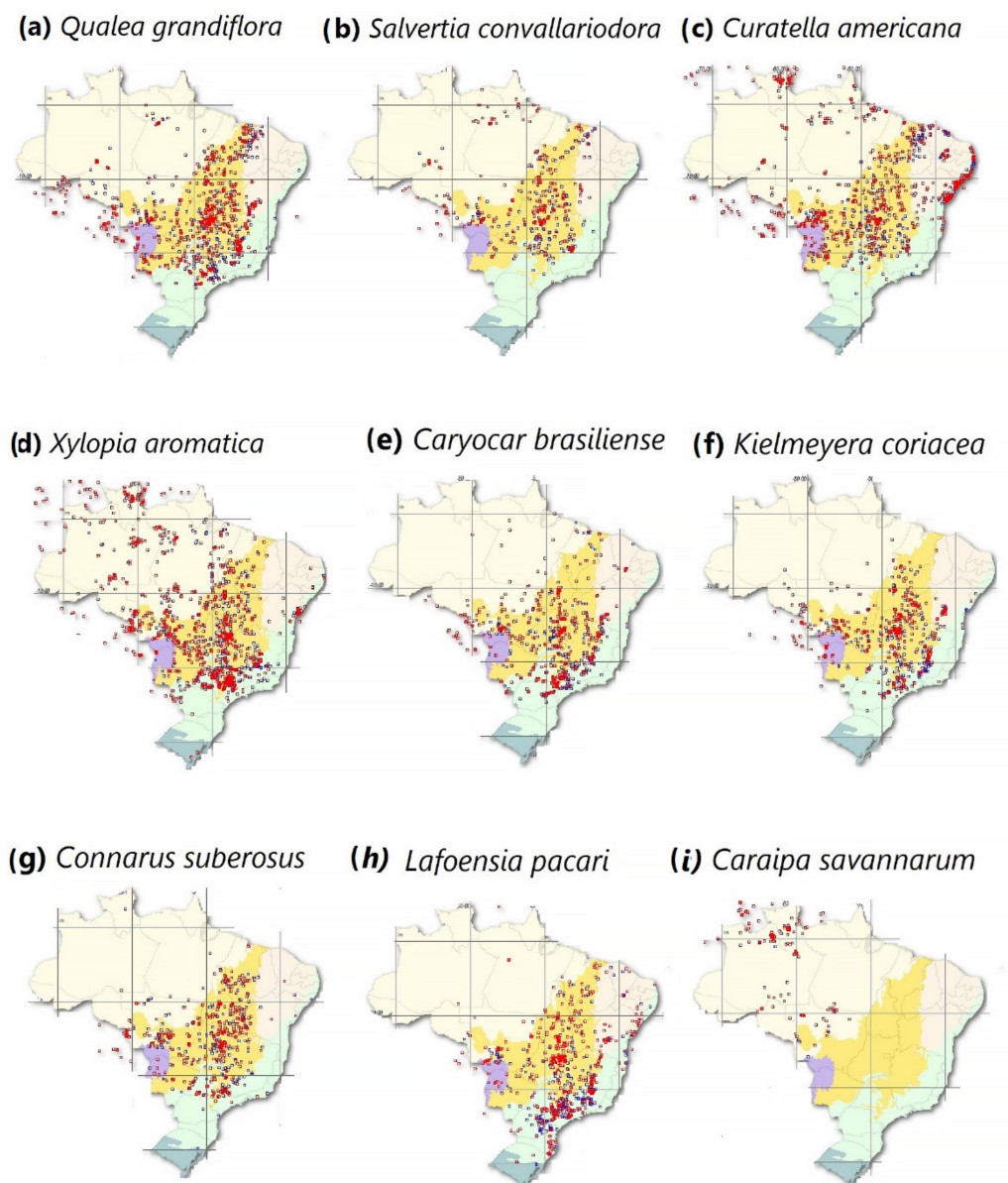

**Figure 1.** The geographical distribution of typical savannah species that are found in the Brazilian Amazon. The points displayed on the maps represent original species geographic coordinates (red) and per occurrence municipality in Brazil (blue). Reprinted with permission from ref. [124]. 2021 speciesLink network.

## 4. Discussion

### 4.1. Deforestation and Forest Degradation Implications to Forest Dependent Communities

Deforestation reduces the livelihoods of forest-dependent communities in the Amazon by abruptly eliminating about 500 million trees and palms each year [16,40,44]. Forest fragmentation increases plant mortality along the edges [58,59]. Local climate change, like increased surface air temperature and decreased humidity, influences the ecophysiology of many species [58]. The physical and biological changes associated with deforestation and forest fragmentation cause widespread loss of animal disperses and pollinators and affect flower, fruit, and seed development [5,35,125].

Selective logging has reduced the supply of forest-dependent communities in the Amazon. However, some organizations argue that selective logging in the Amazon can take place with reduced impacts [126]. These arguments underpin the Brazilian legislation that grants areas of public Amazonian forest to private, selective logging companies through a bidding process [127]. However, severe and significant selective logging affects the forest structure, microclimate, and plant species composition [10,80,82,83,128]. In fact, people living in the Amazon who depend on the forest for income report negative effects of selective logging on NTFP collection in countries such as Brazil [9], Bolivia [129], and Guyana [130]. Similar reports are noted in the tropical forests of Costa Rica, Nicaragua, Cameroon, and Malaysia [10]. Therefore, granting public Amazonian forests for selective logging presents both a social and economic threat to communities extracting NTFPs.

Forest fires affect NTFP extraction activities and the cultivation of native species by destroying vegetation and temporarily or permanently interrupting the supply of forest products [11,88,131]. In addition, forest fires increase the risk of infrastructure damage and forest structure [132], species composition, and microclimate modifications [81,105]. Historical data and future projections indicate increased forest fires in the Amazon region [89,133,134]. Therefore, without forest fire prevention and adaptation plans, livelihoods activities linked to forest products from species intolerant to recurrent fires may be reduced.

### 4.2. The Effects of Climate Change on Plant Species

Droughts in the Amazon region are increasing in intensity and frequency [18,19]. The negative effects of drought on tree mortality and fruiting can last up to two or more years after the water deficit period [135]. However, the species mentioned in Table 2 are found in both deforested and abandoned areas, where the microclimate is drier and may be better adapted to the more frequent droughts predicted in the future for the region [100]. Droughts lead to more forest fires [19,104], and the species listed in Table 2 generally occur in both deforested and abandoned areas that are frequently affected by fires [34,79].

Global warming appears to be a key long-term threat to the Amazonian forest products by changing the geographic distribution of plant species [6,23]. It reduces flowers' exposure to pollinators and alters physiological processes associated with protein, oil, and vegetable fat syntheses [21,111]. Pollination and protein and lipid production are essential for seed formation [31,35].

Increased reports of certain species, such as *C. savannarum* and *Cecropia* [16,122], may be evidence of a regional climatic shift that favors the replacement of typical humid climate-adapted species with typical savannah species [1,50]. In fact, although poorly studied, changes in species composition are considered a slow and gradual process [16,51]. The Amazon region vulnerable to "savannization" is characterized by an open ombrophilous forest type [50,85]. To a large extent, this region is located in the southern Brazilian Amazon forest, encompassing municipalities in Acre, south Amazonas, Rondônia, north Mato Grosso, and southern Pará [13,85]. Therefore, typical savannah species populations, such as those in Figure 1, may increase in the southern and eastern regions of the Brazilian Amazon, where the environment is drier, warmer and where frequent forest fires occur.

### 4.3. Potential Reduction of Native Amazonian Plants and Annual Range of Economic Losses

There will be a reduction of native Amazonian species widely employed by local populations (for example, Table 1) due to environmental changes, both local and globally, in turn leading to decreased supply, diversity, quality, and amount of forest products. This suggests that without the restoration of deforested and degraded forests, the food, well-being, and income obtained by communities from native forest products in the Amazon will be severely impacted (Figure 2). Other human activity effects that alter ecosystem productivity and geographical distribution of plant species in the Amazon are noted, in addition to those reviewed herein, such as extreme flooding, over-exploitation, invasive species, and loss dispersers [136–138]. Species that dominate the forest succession in

deforested and abandoned lands are not economically exploited in the region (as listed in Table 2), with the exception of palms.

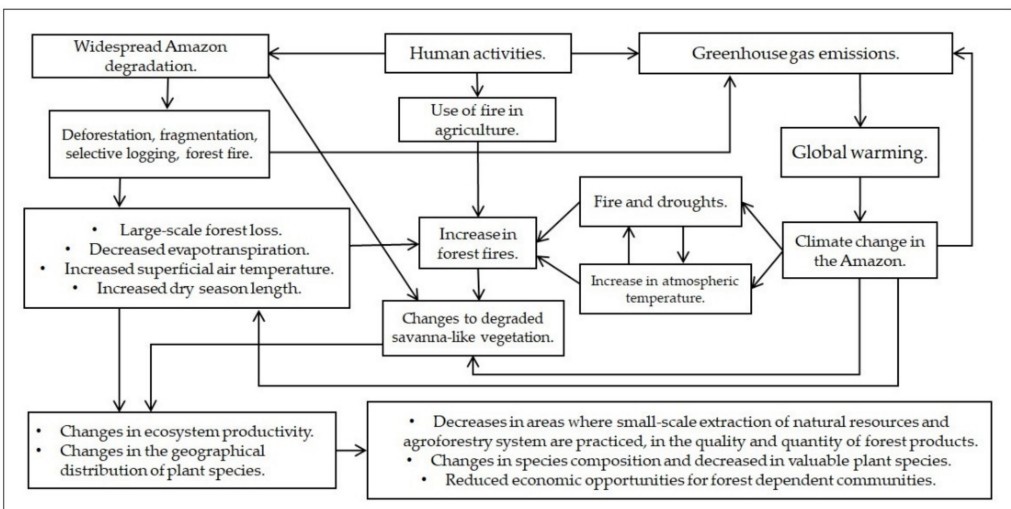

**Figure 2.** Schematic representation of plant responses to human environmental changes and livelihoods effects in forest-dependent communities.

Palm species are currently the most important in the Amazon NTFP economy [76,139,140], comprising the botanical Arecaceae family, and occur widely in undisturbed forests [95]. Since they are resistant to forest fires and extreme droughts, they are also present during different vegetation succession phases in both deforested and abandoned areas [46,81,106]. This indicates that food provisioning and income from palm species may be less impacted by some of the environmental changes discussed in this review [46,81,106]. Therefore, an increase in palm density may indicate a future with more palms in the southern and eastern regions of the Amazon.

Finally, a decline of valuable species has been observed mainly in the eastern, central, and southern regions of the Brazilian Amazon [8–10,25,35,48,64,83]. In fact, the value generated in the Brazilian Amazon from NTFPs is approximately US$2 billion year$^{-1}$ [66,139,141]. After deducting collection and transport costs, Peters et al. (1989) estimated that US$422 ha$^{-1}$ year$^{-1}$ was obtained for NTFP alone [27]. As more environmental products and services are included in estimates of tropical forest value potential, the global average is US$5264 ha$^{-1}$ year$^{-1}$ [28]. Based on these estimates and the 16-year average Amazon deforestation rate (about 1,668,600 hectares per year [29]), the loss of native species' economic importance ranges between US$704,149,200 year$^{-1}$ and US$8,783,510,400 year$^{-1}$.

### 4.4. Potential for the Improvement of Agroforestry Systems Strategies and Advancement of Stakeholder Engagement Approaches

Agroforestry systems combine the management of native and exotic species simultaneously in the same land. For example, native plants highly present in agroforestry systems in Amazon are acai berry (*Euterpe oleracea* Mart. and *E. precatoria*), andiroba (*C. guianensis*), buriti (*Mauritia flexuosa* L.), bacuri (*Platonia insignis* Mart.), cacau (*Theobroma cacao* L.), Brazil nut (*B. excelsa*), tonka bean (*D. odorata*), cupuassu (*Theobroma grandiflorum* (Willd. ex Spreng.) K.Schum.), pupunha (*Bactris gasipaes* Kunth.), and taperebá (*Spondias mombin* L.), while common exotic species are coconut (*Cocos nucifera* L.), coffee (*Coffea arabica* L. and *Coffea canephora* Pierre ex A.Froehner.), palm oil (*Elaeis guineenses* Jacq.), mango (*Mangifera indica* L.), banana (*Musa × paradisiaca* L.), lemon, orange, tangerine (*Citrus* spp.), and black pepper (*Piper nigrum* L.). Agroforestry systems in the Amazon are more productive and biodiverse compared to monocultures and can also include animal management, such as cattle, chickens, ducks, and pigs, without having to deforest down huge forest areas [8,142–144].

Agroforestry system costs and benefits in the Amazon are promising to public and private investments, especially concerning forest restoration. The average investment required for forest restoration through the agroforestry system method ranges between US$2500 and US$7000 per hectare [142,145,146], with internal return rates ranging from 10 to 111%, and payback between 2 and 13 years [145,146]. Income generation with NTFPs obtained from agroforestry systems has varied between US$400 and US$800 ha$^{-1}$ year$^{-1}$ [146]. In fact, native species NTFPs are generating around US$1 billion/year$^{-1}$ in municipalities in the state of Pará, with an estimated economic value of US$32 billion by 2040 if investments in science and technology are made [141]. Furthermore, forest restoration through the agroforestry systems has resulted in three times more aboveground biomass compared to the average in degraded and abandoned areas [44,147], consequently comprising a Reducing Emissions from Deforestation and Forest Degradation (REDD+) strategy [148].

The demand from industrial activities for NTFP is to increase the quality and quantity of raw material to the supply chain [7]. Companies in the cosmetics, food, chemical, and perfume industries are already being supplied with NTFP from Amazonian plants [7,76]. The technological and scientific challenges, however, are significant. For example, only 15% of municipalities in the Brazilian Amazon have the industrial infrastructure to produce raw materials, such as fats, oils, and pulp, for the national and global markets [8]. In addition, technical assistance investments are crucial for businesses and people who depend on NTFP in the Amazon, but the number of establishments that receive technical assistance ranges between 1 and 22% in the different regions [141]. Therefore, science and technology investments are considered promising in implementing agroforestry systems recovering deforested and degraded areas, which could engage companies that use NTPF due to supply chain advantages.

Science and technology investments are still low in the Amazon. For example, the Brazilian National Bank for Economic and Social Development (BNDES) is the largest public resource investor for the development of NTFP supply chains in the Brazilian Amazon, investing an average of US$10 million year$^{-1}$ [149]. The largest private investor is located in the cosmetic company, averaging US$50 million year$^{-1}$ [76]. In addition, deforested and abandoned lands are rarely being restored, with native species presenting economic value [65], and NTFPs are rapidly reducing due to deforestation, forest degradation, and climate change [8]. Therefore, new investments are essential for people, businesses, and sustainable value-chains in the Amazon, which will strengthen entrepreneurship, innovations, and startups [7,13,150–152].

The species lists presented herein can aid in the species selection phase for agroforestry system implementation and can be used as references for stakeholders. Both species lists contain empirical references to changes in species composition due to human activities, which can be revised and enhanced in future studies. Furthermore, other species that may increase in agroforestry systems in the Amazon comprise those typically employed as NTFP in the savannah vegetation region and present in the Amazon, such as the araticum (*Annona crassiflora* Mart.), cashew (*Anacardium occidentale* L.), macauba (*Acrocomia aculeata* (Jacq.) Lodd. ex Mart.) e pequi (*Caryocar brasiliense* Cambess.). A research agenda on species increasing in degraded land can augment the social and economic importance of these plant species.

## 5. Conclusions

Environmental changes result in decreased ecosystem productivity and geographical distribution of plant species in the Amazon region and lead to decreased plant diversity, biomass, and wood density, as well as decreased amounts and quality of forest products. Deforestation and global warming are the largest threats to the economic importance of plants in the Amazon region, both in the short and long term. This will result in diminishing economic opportunities for forest-dependent communities, as well as accelerated biodiversity loss.

Certain less economically exploited native species, such as *Cecropia* spp. and *Vismia* spp., as well as typical savannah species that occur in the parts of the Amazon region, are better adapted to droughts and forest fires. The change in plant species composition in degraded areas occurs mainly in the southern and eastern regions of the Brazilian Amazon, where the local climate is naturally drier and the dry season is longer, due to which there are more frequent forest fires. Thus, the populations of less economically exploited and savannah species might increase in these regions.

The decrease in the economic importance of plants occurs at different temporal scales. Over 1 billion native trees and palms are lost every two years in the Amazon forest, causing economic losses estimated between US$1–17 billion. Biodiversity loss can be abrupt and both temporary or persistent for over 20 years. A lack of efficient environmental change mitigation and adaptation plans may lead to continuous decreases in the economic importance of native plant species.

Finally, to maintain the forest's social and economic importance, native plant species with economic value that are declining must be planted, as the reviewed studies indicate that their populations are decreasing without forest restoration efforts. The native plant species that are increasing their population are more adapted to environmental change and may be employed in forest restoration, although, on the other hand, they are still underused in the region's economy. New studies are important to understand how typical savannah species are affected by environmental changes currently taking place in the Amazon.

**Author Contributions:** Conceptualization, D.O.B. and C.A.N.; methodology, D.O.B.; validation, L.E.S.B. and C.A.N.; formal analysis, D.O.B.; investigation, D.O.B.; writing—original draft preparation, D.O.B.; writing—review and editing, L.E.S.B. and C.A.N.; supervision, C.A.N.; funding acquisition, D.O.B. All authors have read and agreed to the published version of the manuscript.

**Funding:** D.O.B. was supported by the National Council for Scientific and Technological Development (Conselho Nacional de Desenvolvimento Científico e Tecnológico—CNPq) for a doctoral grant (Project: 870001/2011-6, Process: 141882/2018-2, valid between 5 January 2018 and 12 April 2018), and by both the Fundo Brasileiro para a Biodiversidade and HUMANIZE for Funbio grant (Conservando o Futuro n° 035/2019).

**Data Availability Statement:** Not applicable.

**Acknowledgments:** D.O.B. thanks the postgraduate program in Earth System Science (Pós-Graduação Ciência Sistema Terrestre—PGCST) and the National Institute for Space Research (Instituto Nacional de Pesquisas Espaciais—INPE) for their support throughout the course of this study. We also want to thank the reviewers for their helpful comments on the first version of this manuscript.

**Conflicts of Interest:** The authors declare no conflict of interest.

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
