# Peer review of "The Effects of Environmental Changes on Plant Species and Forest Dependent Communities in the Amazon Region"

_forests, doi:10.3390/f13030466_

Round 1

Reviewer 1 Report

Title of the manuscript: Anthropogenic drivers of environmental change reduce the economic potential of plant species in the Amazon

This study aimed to investigate how environmental changes affect the productivity and geographical distribution of plant species in the largest rainforest in the world in the Amazon. IPCC reports and scientific studies are the main data sources for analysis to achieve this goal. This paper has potentials to be published in the journal of Forests, though some issues need to be clarified before proceeding further. To improve the structure of the paper, the authors should address the following comments:

Keywords

  1. There are too many keywords for this paper. The keywords are usually limited to 5.

Materials and Methods

  1. This part should not be divided into subsection if there is only one subsection. Thus, the subtitle of "2.1 Data collection and analysis" should be deleted.
  2. This section is weak. The authors should introduce data sources in more details. For example, the authors should explain how many scientific studies are included in this analysis? Why choose these studies? What is the criteria for literature selection? More importantly, what method is used for this analysis? i.e. how is the data analyzed and why use this kind of method to analyze data?

Results

  1. Actually, the content in this section does not match the title of manuscript well. The manuscript title has two keywords, anthropogenic drivers and economic potential. However, the results presented here are mainly the effects of environmental changes (deforestation, forest fragmentation, forest fire, droughts, global warming) on plant species (seeds and fruit production and species composition, seeds and fruit diversity and density, mortality of plant species, changes in vegetation, changes to savanna vegetation). Economic potential of plant species is quite different concept with that of plant species. The results are not focused on anthropogenic drivers and economic potential as stated in the title. The authors should be very clear about the key research question of this article and reorganize the Results part, or to consider how to revise the title.

Author Response

Summary of reviewer comments & status of the response: Manuscript ID: forests-1559976

Keywords: There are too many keywords for this paper. The keywords are usually limited to 5.

Response: Done. The keywords kept were: agroforestry system; climate change; deforestation; forest degradation; non-timber forest products.

Materials and Methods: This section is weak. The authors should introduce data sources in more details. For example, the authors should explain how many scientific studies are included in this analysis? Why choose these studies? What is the criteria for literature selection? More importantly, what method is used for this analysis? i.e. how is the data analyzed and why use this kind of method to analyze data?

Response: Done. We have prepared substantial changes in the methodology section to better explain the manuscript. We have elaborated five new subsections on materials and methods: 2.1. Criteria for Literature Selection; 2.1.1. Building tables to demonstrate reducing and increasing populations of plant species in deforested Amazon lands; 2.1.2. Maps concerning typical Amazon savannah species; 2.1.3. Number of destroyed trees and palms and potential economic losses; 2.1.4. Potential for agroforestry system strategy improvement.

Results: Actually, the content in this section does not match the title of manuscript well. […] Economic potential of plant species is quite different concept with that of plant species. […]The authors should be very clear about the key research question of this article and reorganize the Results part, or to consider how to revise the title.

Response: Done. The title has been revised to “The Effects of Environmental Changes on Plant Species and Forest Dependent Communities in the Amazon Region”.

Reviewer 2 Report

This study is very important because it addresses issues related to the impact of anthropogenic activities on forest species and especially on flora and fauna, bringing consequences not only of biodiversity loss but also considerable economic losses.

Perhaps a more detailed description could be made on the loss of forest species and biodiversity due to non-anthropogenic activities and that somehow are complemented with the losses due to anthropogenic activities, in the same way, it is considered that more depth should be given to concrete and viable proposals to somehow mitigate these problems, involving social, economic, environmental and cultural issues.

Author Response

Dear Reviewer, We appreciate your time and energy in providing comments and suggestions. Your recommendations are important, but we have some scope limitation to implement them in the manuscript. Please see specific answers below. We are also sending the attachment with the general answers.

Thank you for your time and consideration.

Sincerely

Comments and Suggestions for Authors: Perhaps a more detailed description could be made on the loss of forest species and biodiversity due to non-anthropogenic activities and that somehow are complemented with the losses due to anthropogenic activities […].

Response: This recommendation has been important, but it has not been done. We felt that adding the scientific literature on non-anthropogenic activities could make the manuscript more extensive. Finally, our objective has been to review the literature linked to human activities.

Comments and Suggestions for Authors: […] in the same way, it is considered that more depth should be given to concrete and viable proposals to somehow mitigate these problems, involving social, economic, environmental and cultural issues.

Response: We present the literature on agroforestry systems as a way to reduce the negative impacts of environmental changes. This strategy involves social, economic, environmental and cultural issues. However, there is a need to present other actions, for example, elimination of deforestation and implementation of the Paris Agreement. Presenting more options would require a greater amount of literature review. This can be analyzed in future manuscript

Reviewer 3 Report

Rating the Manuscript

The article is devoted to an interesting and relevant topic. The article has one and a half hundred references, which indicates that the authors are well acquainted with the research topic, have own findings.

  • Originality/Novelty: In my opinion, the authors should better describe the scientific novelty of the article, make it more clearly. Because the large number of references makes it difficult to identify the author's achievements, developed in the article under consideration.
  • Significance: In Results chapter authors write about deforestation, forest degradation, forest fires impact etc., using recent scientific literature. Therefore, the authors' opinions are consistent with the available data and recent discourse.
  • Quality of Presentation: The article is written in an appropriate way, its structure is correct. The data are presented as tables, figures and maps. Maps of the geographical distribution of typical savannah species that are found in the Brazilian Amazon are presented for 9 species.
  • Scientific Soundness: authors do not disclose methods and criteria, they use for data collection, analysis and projection. Hypotheses and speculations should be identified more carefully. Hence, Section 2 of the article needs significant refinement. In the Abstract and Conclusions Authors write: “Over 1 billion native trees and palms are lost every two years in the Amazon forest, causing economic losses estimated between US$ 1–14 billion.” (lines 1-20; 499-500). Relevant substantiation is needed for such a statement.
  • Interest to the Readers: the conclusions are interesting for the readership of the Journal: they are multidisciplinary and combine botany, biology, forestry, economy and sustainability science. Authors try to link ecological and economic values. It is in line with the goal of the Special issue on "The Dynamic Interaction between People and Forest Ecosystems".
  • Overall Merit: Publication of the article will provide overall benefit to scientists and practitioners, It will give an overview of recent trends, discourse and knowledge and highlights future strategies development. 
  • English Level: the English language is appropriate and understandable, but additional reading is needed for minor corrections (for example: “several economically plant species”, line 237-238 etc.).

Special comments:

  • Article title: In my opinion, the title of the article needs to be adjusted, because the economic potential of plant species in the Amazon is reduced due to non-proper human activity but not because of the drivers as such.
  • Goal and hypothesis of the article not clearly formulated
  • Methods and materials should be better explained. To forecast future trends of the processes quantitative methods could be used
  • Which methods / criteria were used to build tables S1 and S2?
  • Anthropogenic drivers of environmental change, explicitly mentioned in the title, should be listed/highlighted on the Fig. 2 (block Human activities).
  • Figure 2: The figure needs to be rethought. There must be a clearer logic of representation. For instance: The first block of the figure is named “Anthropogenic drivers of environmental change reduce the economic potential of plant species in the Amazon”. It looks like a statement, not the name of the figure block. Anthropogenic drivers are not specified. One of the next blocks is “Human activities”. This block is not linked to the first, abovementioned block while connections between some other blocks are shown twice, for example “Climate change in the Amazon” and “Changes to degraded savanna-like vegetation”.. Among activities one can see only “Use of fire in agriculture”. What about other activities, which reduce the economic potential of plant species in the Amazon?
  • Title of the Fig. 2; I do not agree that this is THEORETICAL framework.
  • There is a typo: Deforestation rate is measured in hectares, not in ha-1 (line 420).
  • References: should be prepared according to the journal requirements (there are some typos, for example sources 139, 141 etc.)
  • The article will look more logical when authors provide links between sections / subsections of the article (e.g. 4.4)
  • Authors consider Table S2 and then Table S1
  • Article will look better, if Authors explain the concepts “extractive families” and “selective logging” in the research context.

Author Response

Dear Reviewer, We appreciate your time and energy in providing comments and suggestions. Your recommendations are important, but we have some scope limitation to implement them in the manuscript. Please see specific answers below. We are also sending the attachment with the general answers. 

Thank you for your time and consideration.

Sincerely

Comments and Suggestions for Authors: Originality/Novelty: In my opinion, the authors should better describe the scientific novelty of the article, make it more clearly. Because the large number of references makes it difficult to identify the author's achievements, developed in the article under consideration.

Response: The manuscript is typically a review of existing literature, although there are original calculations as well. The main novelty was broadly characterize environmental changes in the Amazon to assess why the livelihoods of people who depend on forest products are increasingly threatened. The paragraph where the purpose of the manuscript is presented was modified in an attempt to make it clearer for readers.

Comments and Suggestions for Authors: Scientific Soundness: authors do not disclose methods and criteria, they use for data collection, analysis and projection. Hypotheses and speculations should be identified more carefully. Hence, Section 2 of the article needs significant refinement. In the Abstract and Conclusions Authors write: “Over 1 billion native trees and palms are lost every two years in the Amazon forest, causing economic losses estimated between US$ 1–14 billion.” (lines 1-20; 499-500). Relevant substantiation is needed for such a statement.

Response: Done. The content of the sections is now better explained in the methodology section. We have prepared five new subsections: 2.1. Criteria for Literature Selection; 2.1.1. Building tables to demonstrate reducing and increasing populations of plant species in deforested Amazon lands; 2.1.2. Maps concerning typical Amazon savannah species; 2.1.3. Number of destroyed trees and palms and potential economic losses; 2.1.4. Potential for agroforestry system strategy improvement.

Comments and Suggestions for Authors: Article title: In my opinion, the title of the article needs to be adjusted, because the economic potential of plant species in the Amazon is reduced due to non-proper human activity but not because of the drivers as such.

Response: Done. The title has been changed to “The Effects of Environmental Changes on Plant Species and Forest Dependent Communities in the Amazon Region”.

Comments and Suggestions for Authors: Goal and hypothesis of the article not clearly formulated.

Response: Done.

Comments and Suggestions for Authors: Methods and materials should be better explained.

Response: Done.

Comments and Suggestions for Authors: To forecast future trends of the processes quantitative methods could be used.

Response: This recommendation was not fully understood. Despite this, we mainly have used quantitative information to develop the manuscript.

Comments and Suggestions for Authors: Which methods / criteria were used to build tables S1 and S2?

Response: Done. We created the "Building tables to demonstrate reducing and increasing populations of plant species in deforested Amazon lands" section in the methodology to explain the tables.

Comments and Suggestions for Authors: Anthropogenic drivers of environmental change, explicitly mentioned in the title, should be listed/highlighted on the Fig. 2 (block Human activities). Figure 2: The figure needs to be rethought. There must be a clearer logic of representation. For instance: The first block of the figure is named “Anthropogenic drivers of environmental change reduce the economic potential of plant species in the Amazon”. It looks like a statement, not the name of the figure block. Anthropogenic drivers are not specified. One of the next blocks is “Human activities”. This block is not linked to the first, abovementioned block while connections between some other blocks are shown twice, for example “Climate change in the Amazon” and “Changes to degraded savanna-like vegetation”.. Among activities one can see only “Use of fire in agriculture”. What about other activities, which reduce the economic potential of plant species in the Amazon?

Response: We made a few modifications to the schematic in Figure 2 and a substantial change in the title of the article.

Comments and Suggestions for Authors: Title of the Fig. 2; I do not agree that this is THEORETICAL framework.

Response: The figure caption has been changed to "Schematic representation of plant responses to human environmental changes and livelihoods effects in forest dependent communities."

Comments and Suggestions for Authors: There is a typo: Deforestation rate is measured in hectares, not in ha-1

Response: Revised.

Comments and Suggestions for Authors: References: should be prepared according to the journal requirements (there are some typos, for example sources 139, 141 etc.)

Response: Done. The references have been revised to meet the journal's requirements.

Comments and Suggestions for Authors: The article will look more logical when authors provide links between sections / subsections of the article (e.g. 4.4)

Response: Done. The content of the sections is now better explained in the methodology section.

Comments and Suggestions for Authors: Authors consider Table S2 and then Table S1

Response: This has been changed. In addition, the tables are now presented in the results section, unlike the first version of the manuscript that were in the Supplementary Materials.

Comments and Suggestions for Authors: Article will look better, if Authors explain the concepts “extractive families” and “selective logging” in the research context.

Response: Done.

Round 2

Reviewer 1 Report

I have read the revised version of the paper, I think my concern has been addressed.

Author Response

Dear Reviewer,

Thank you for your email dated 1 March 2022.

The best

Reviewer 3 Report

Although the authors have done a lot of work on the article, there are still some insufficiently taken into account comments, which will improve the article, in particular:

  • Authors should explain which bibliometric database use for their scientific search: were they peer-review journals let say from Scopus, WoS etc. or national databases or grey literature, which soft they used for the search, if any, what period of time it covers.
  • Authors should explain better which criteria /rules they used for the literature search. Paragraph 2.1. Criteria for Literature Selection still does not include them .
  • The number US$ 1–14 billion (lines 20 and 574) remains without relevant justification or reference

Author Response

Dear Reviewer,

Thank you for your email dated 3 March 2022. We would like to thank you again for their time and energy in providing comments, and helpful suggestions on the manuscript. In addressing their comments, we feel we have significantly improved the clarity of the manuscript, and hope you will now find it to be a better fit to Forests readers. Please see our responses below.

Yours sincerely,

Diego Oliveira Brandão, on behalf of my coauthors.

Comments and Suggestions for Authors:

Authors should explain which bibliometric database use for their scientific search: were they peer-review journals let say from Scopus, WoS etc. or national databases or grey literature, which soft they used for the search, if any, what period of time it covers.

Authors should explain better which criteria /rules they used for the literature search. Paragraph 2.1. Criteria for Literature Selection still does not include them.

Response:

Forests' general considerations of review manuscripts require the front matter, literature review sections and the back matter. These requirements have been implemented. The review manuscript references clearly indicate the origins of the literature, which are mostly from peer-review journals. We believe that referencing Scopus, WoS, national database in materials and methods will not provide more or better information than has already been written.

The “Scientific writing = thinking in words” guide demonstrates that “The structure of the review: The format and the layout from this point are seldom prescribed in detail by journals and are certainly not as rigidly constrained as are those in a research article. They can vary with the topic and its scope and give the opportunity to develop the layout more freely” (page 96). Recent review manuscripts were published without providing information on the bibliometric database (for example: Ribeiro et al in Forests 2022, 13, 386. https://doi.org/10.3390/f13030386), while other reviews used the bibliometric database (for example: Di Cori et al in Forests 2022, 13, 362. https://doi.org/10.3390/f13030362). In fact, other reviewers of this manuscript did not request information on the bibliometric database. Therefore, we understand that a bibliometric database, rules, period of time, and software in the materials and methods are not required to understand this review manuscript.

Comments and Suggestions for Authors:

The number US$ 1–14 billion (lines 20 and 574) remains without relevant justification or reference.

Response

Done. We have made substantial changes.

Subsection "2.1.3. Number of destroyed trees and palms and potential economic losses. The number of trees and palms was estimated using information on the average number of stems per hectare and deforested area from 1988 to 2020. Economic losses are presented alongside the net annual revenues from NTFP and the global estimates regarding the value of ecosystems and their services in monetary units." has been replaced by " 2.1.3. Annual Range of Economic Importance Losses Caused by Deforestation We employed peer-review articles to estimate the annual range of economic losses caused by Amazon deforestation [30,86,142]. The net annual revenues from NTFP were used to estimate a minor set of ecosystem services losses due to deforestation [142]. We considered the annual revenues from NTFP in US$ 422 ha-1 year-1, according to estimates for the Peruvian Amazon [142]. The average monetary values from tropical forests were used to estimate a major set of ecosystem services losses by deforestation [30]. We considered the average monetary value estimated as US$ 5264 ha-1 year-1, according to calculations based on a total of 17 types of ecosystem services, including provisioning services (i.e., food, water, raw materials, genetic and medicinal resources), regulating services (i.e., air quality regulation, climate regulation, erosion prevention, biological control), habitat services (i.e., nursery service and genetic diversity), and cultural services (i.e., recreation) [30]. The annual average Amazon region deforestation estimated as in 16,686 km2 during 2002–2018 was used [86]. Thus, we present and discuss an annual range of losses of economic importance caused by deforestation in the Amazon.”.